# *In Vitro* synergy of Farnesyltransferase inhibitors in combination with colistin against ESKAPE bacteria

**Marian Klose, Lea Weber, Hagen Sjard Bachmann** [ID]*

Faculty of Health, Institute of Pharmacology and Toxicology, Centre for Biomedical Education and Research (ZBAF), School of Medicine, Witten/Herdecke University, Witten, Germany

* Hagen.Bachmann@uni-wh.de

## Abstract

The emergence of antibiotic resistance continues to pose a significant global challenge. Drug repurposing, wherein existing therapeutics are evaluated for new applications, offers a promising strategy to address this issue. Farnesyltransferase inhibitors (FTIs), initially developed for cancer therapy, have demonstrated antimicrobial activity against several gram-positive bacteria. This study investigates their activity in combination with colistin against gram-positive and gram-negative bacteria. We focus on key ESKAPE (*Enterococcus faecium*, *Staphylococcus aureus*, *Klebsiella pneumoniae*, *Acinetobacter baumannii*, *Pseudomonas aeruginosa*, and *Enterobacter* species) pathogens while incorporating additional bacterial strains to provide a comprehensive understanding of differential responses and potential dose-dependent synergistic effects. Antimicrobial susceptibility was assessed using broth microdilution, while synergy was evaluated through checkerboard, time-kill, and growth kinetics assays. When combined with sub-inhibitory colistin, FTIs inhibited gram-negative bacterial growth. Tipifarnib exhibited more potent antimicrobial activity against gram-negative strains than lonafarnib. Peptidomimetic FTIs, B581 and FTI-277, inhibited gram-negative bacteria in combination with colistin but had no effect on the gram-positive strains tested. In contrast, alpha-hydroxy farnesyl phosphonic acid, an FPP analog, and bempedoic acid, targeting the mevalonate pathway, showed no antibacterial activity. In addition to their known inhibition of gram-positive bacteria, FTIs exhibited efficacy against gram-negative bacteria, including colistin-resistant *Enterobacter cloacae* subsp. *cloacae*, when combined with sub-inhibitory colistin. This might be due to a mechanism distinct from their eukaryotic targets, potentially involving the disruption of multiple biosynthetic pathways. Future studies will focus on elucidating these mechanisms of FTIs and exploring the therapeutic potential of FTI/colistin combinations against ESKAPE and other multidrug-resistant pathogens.

**Data availability statement:** All relevant data are within the manuscript and its Supporting Information files.

**Funding:** The author(s) received no specific funding for this work.

**Competing interests:** The authors have declared that no competing interests exist.

## Importance

Antibiotic resistance is a growing threat to global health, necessitating innovative strategies to combat resistant pathogens. This study highlights the potential of farnesyltransferase inhibitors (FTIs), originally developed for cancer therapy, as antimicrobials against key pathogenic bacteria. When combined with colistin, FTIs demonstrated significant activity against gram-negative bacteria. Notably, this combination enabled FTIs to exert antimicrobial activity even against colistin-resistant bacteria. Although the exact mechanism of action remains to be elucidated, this combination may offer a valuable therapeutic option against bacteria that have developed increased resistance to colistin, addressing a critical need in the treatment of multidrug-resistant pathogens.

## Introduction

The emergence of antibiotic resistance is a steadily growing concern [1]. At the same time, research into novel antimicrobial agents is proving difficult and cost-intensive [2]. If current trends continue, it is estimated that antibiotic-resistant bacteria alone could cause over 10 million deaths per year by 2050 [3]. One way to at least partially address these problems is to test already established drugs of other medical areas for possible antimicrobial effects (drug repurposing). This was successfully conducted by using statins, which are lipid-modifying agents due to their competitive binding to 3-hydroxy-3-methylglutaryl coenzyme A (HMG-CoA) reductase. One of the multiple pleiotropic properties also revealed antimicrobial potential in addition to its main effect [4].

Likewise, we were able to show that inhibitors of farnesyltransferase (FTase), which are used as cancer therapeutics, also exhibit antimicrobial effects [5]. However, antimicrobial effects of farnesyltransferase inhibitors (FTIs) have so far only been demonstrated in gram-positive bacteria (Fig 1A). Given that the outer membrane (OM) may limit the activity of FTIs, colistin, which binds to Lipid A in the lipopolysaccharide layer, thereby disrupting the OM's integrity and increasing its permeability, could serve as a means to expand their spectrum of activity to gram-negative bacteria [8,9].

Similar studies have already been conducted with other drugs. For example, Huang, Zhu [10] were able to show that increased intracellular accumulation of flavomycin can be achieved with simultaneous administration of sub-inhibitory colistin. This led to a significant reduction in the amount of colistin required to reduce the bacterial cell count. While these findings highlight the potential of combination therapies, the antimicrobial effects of FTIs have not been broadly explored across all ESKAPE pathogens in a combinational approach with colistin. The ESKAPE panel, consisting of *Enterococcus faecium*, *Staphylococcus aureus*, *Klebsiella pneumoniae*, *Acinetobacter baumannii*, *Pseudomonas aeruginosa*, and *Enterobacter* species, represents a major global health threat due to their ability to "escape" the effects of many current antibiotics [11,12]. These pathogens are not only responsible for a significant proportion of nosocomial infections but also exhibit a high prevalence of multidrug resistance, complicating treatment options [13]. Their virulence and adaptive mechanisms enable rapid genomic adaptation and resistance development, making it even more critical to discover new treatment options [14,15].

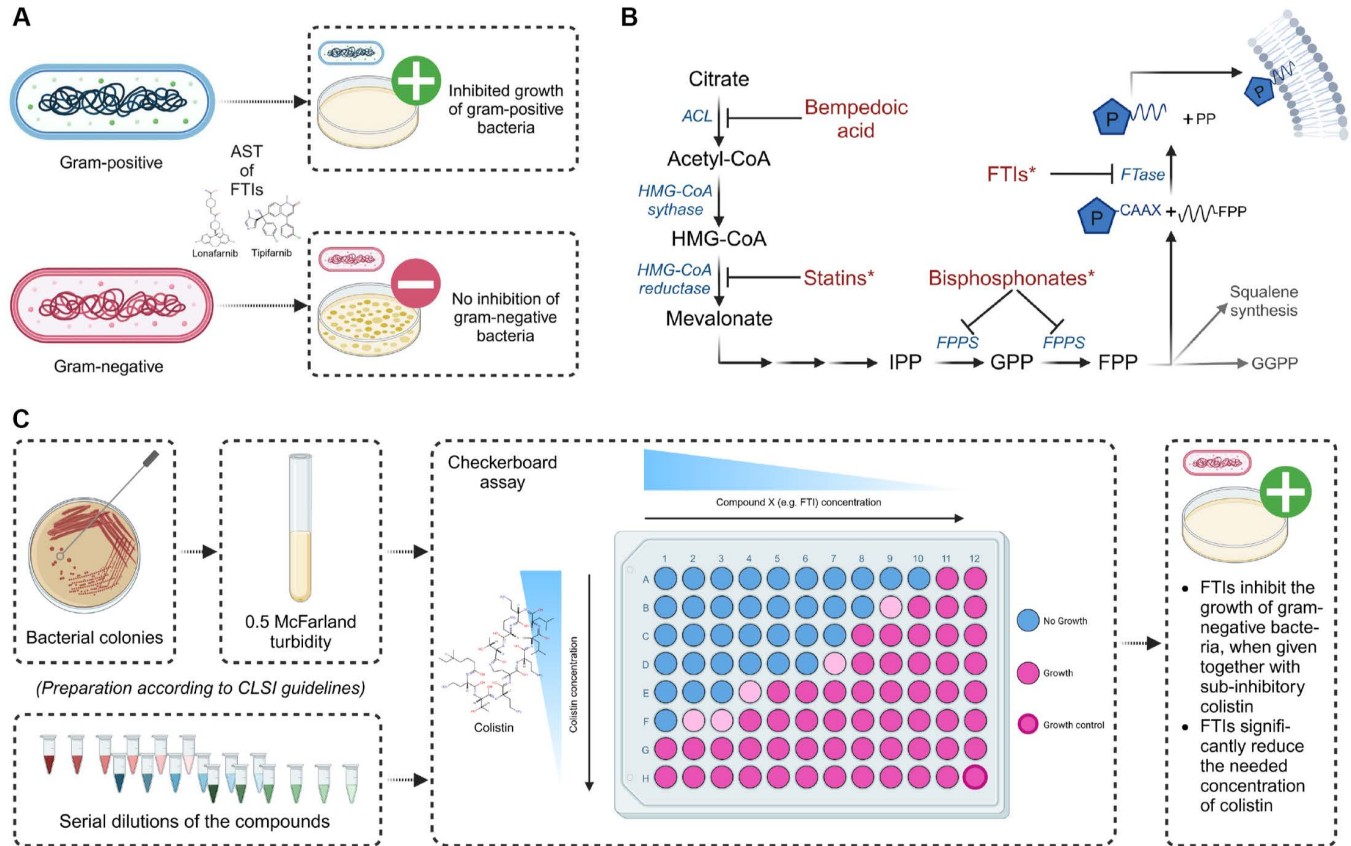

**Fig 1. Antimicrobial activity of farnesyltransferase inhibitors.** (A) shortly summarizes the antimicrobial activity of the farnesyltransferase inhibitors (FTIs) lonafarnib and tipifarnib, which demonstrated efficacy against gram-positive bacteria, including pathogenic *Staphylococcus* species. However, no activity was observed against gram-negative bacteria [5]. (B) gives an overview of the used drugs and their corresponding targets in eukaryotic cells. Drugs that show antimicrobial potential are marked (*): Statins, Bisphosphonates [6,7], and FTIs. (C) highlights the primary objective of this study: expanding the spectrum of FTIs to gram-negative bacteria by combining them with sub-inhibitory concentrations of colistin. This approach enables FTIs to exhibit antimicrobial activity against several gram-negative bacteria when co-administered with sub-inhibitory concentrations of colistin, while also potentially reducing the colistin dose required for growth inhibition, offering a novel strategy to combat antimicrobial resistance. Abbreviations: ACL, ATP-citrate lyase; AST, antimicrobial susceptibility testing; FPP, farnesyl pyrophosphate; FPPS, farnesyl pyrophosphate synthase; FTase, farnesyl-transferase; GGPP, geranylgeranyl pyrophosphate; GPP, geranyl pyrophosphate; HMG-CoA, 3-hydroxy-3-methylglutaryl-coenzyme A; IPP, isopentenyl pyrophosphate; PP, pyrophosphate. Created in BioRender.

In addition to pathogenic bacteria, interactions with commensal bacterial species are relevant when assessing novel antimicrobial approaches. The balance of microbial communities, such as those in the gut or on the skin, plays a critical role in maintaining host physiology [16]. Disrupting this balance through antimicrobial interventions could potentially promote the overgrowth of opportunistic pathogens [17], highlighting the importance of understanding antimicrobial effects beyond traditional pathogenic targets.

The specific objectives of this study are as follows: (i) to evaluate the effects of the non-peptidomimetic FTIs, lonafarnib and tipifarnib, on gram-positive and gram-negative bacteria when co-administered with colistin at sub-inhibitory concentrations; (ii) to assess the effects of the peptidomimetic FTIs, B581 and FTI-277, under similar conditions; and (iii) to investigate the impacts of alpha-hydroxy farnesyl phosphonic acid, a non-hydrolyzable analog of FPP, and (iv) bempedoic acid, an adenosine triphosphate citrate lyase inhibitor, under the same conditions, to better contextualize the effects of

the aforementioned substances on bacterial behavior. Unlike the previously mentioned substances, which were primarily developed for the treatment of carcinomas, bempedoic acid is a lipid-lowering agent prescribed to a broader patient population. Notably, bempedoic acid and lonafarnib share the advantage of already being approved for clinical use [18,19], with bempedoic acid targeting the cholesterol biosynthesis pathway upstream of statins. Fig 1B provides an overview of these substances and their established eukaryotic targets. To achieve these aims, the checkerboard assay was employed to determine potential synergistic interactions. In addition, we extended the current checkerboard method by incorporating the viability agent resazurin and analyzing the results using fluorescence measurement. Moreover, dose-dependent interactions were systematically assessed, as illustrated in Fig 1C.

## Materials and methods

### Chemicals

Colistin was purchased as colistin sulfate from Sigma-Aldrich (Sigma Aldrich, MO, USA). All additional reference antibiotics were purchased from Roth (Carl Roth, Karlsruhe, Germany). Bempedoic acid (ETC-1002, abbreviated as BPD in this study), FTI-277 (as FTI-277 HCl), lonafarnib and tipifarnib were all purchased from SelleckChem (SelleckChem, Munich, Germany). B581 was purchased from Enzo Life Sciences (Enzo Biochem, Inc., NY, USA), while alpha-hydroxy farnesyl phosphonic acid (αHFP) was purchased from Cayman Chemical (Cayman Chemical, MI, USA). Colistin was dissolved in water and αHFP was dissolved in ethanol. All other substances were dissolved in DMSO.

### Bacterial strains

All bacterial strains were purchased from Leibniz-Institute DSMZ-German Collection of Microorganisms and Cell Cultures. Table 1 lists all used strains alphabetically, giving further information such as the possible affiliation to the ESKAPE group.

### Antimicrobial susceptibility testing

Broth microdilution assay was used to determine minimal inhibitory concentrations (MICs) of each substance, following CLSI (Clinical and Laboratory Standards Institute) guidelines [22]. For evaluation, resazurin (Sigma Aldrich) was used to indicate bacterial growth [23]. Resazurin is reduced to fluorescent resorufin by viable cells, likely via NADPH dehydrogenase [24]. An Infinite M Plex microplate reader (Tecan Trading AG, Switzerland) measured the fluorescence intensity (excitation at 540 nm, emission at 590 nm). Effective inhibition is assumed if the relative fluorescence intensity (RFI) in

Table 1. Alphabetical summary of bacterial strains used in this study.

| Bacterial strain | Abbreviation | ATCC | DSM | ESKAPE[a] |
|---|---|---|---|---|
| *Acinetobacter baumannii* | *A. baumannii* | 19606 | 30007 | • |
| *Bacillus subtilis* strain 168 | *B. subtilis* | – | 23778 | |
| *Enterobacter cloacae* subsp. *cloacae* | *E. cloacae* | 13047 | 30054 | • |
| *Enterococcus faecium* | *E. faecium* | 19434 | 20477 | • |
| *Escherichia coli* BW25133 | *E. coli* | – | 27469 | |
| *Klebsiella pneumoniae* subsp. *pneumoniae* | *K. pneumoniae* | 13883 | 30104 | • |
| *Pseudomonas paraeruginosa*[b] | *P. paraeruginosa* | 9027 | 1128 | • |
| *Staphylococcus aureus* | *S. aureus* | 25923 | 1104 | • |
| Methicillin-resistant *Staphylococcus aureus* | MRSA | 33592 | 11729 | • |
| *Staphylococcus epidermidis* | *S. epidermidis* | 12228 | 1798 | |

[a]ESKAPE organisms are indicated with a dot (•).

[b]The *Pseudomonas* strain ATCC 9027 (DSM 1128), previously classified as *Pseudomonas aeruginosa*, has been reclassified as *Pseudomonas paraeruginosa* based on recent phylogenetic analyses [20,21].

all samples is reduced by more than 90% compared to the growth control at a certain concentration, in addition to the visible absence of growth. Minimal bactericidal concentrations (MBCs) were determined immediately after MIC values by plating 3 µl on agar plates without antibacterial agent. Mueller Hinton medium (Carl Roth, Karlsruhe, Germany) was used throughout the experiments, as broth (MHB) or agar (MHA) with 18 g/l agar. The medium was cation-adjusted [25].

## Checkerboard assay

In order to evaluate the synergistic effects of the individual substances with colistin, a modified checkerboard assay was used, as described by Bellio, Fagnani [26]. Briefly, a 96-well microplate is divided into two equal parts. This allows two checkerboard assays to be tested on one plate. The dilution scheme and the procedure in general are shown in Fig 2A. Duplicate dilution series for colistin (ranging from 0.008 µg/ml to 0.5 µg/ml for all bacteria except *E. cloacae*, where concentrations ranged from 0.06 µg/ml to 4.0 µg/ml) and one of each of the other drugs tested (ranging from 7.8 µM to 125.0 µM) were mixed in cation-adjusted MHB. Each well contained 0.2 ml of each antimicrobial combination or broth control. The final bacterial concentration after inoculation was $5 \times 10^5$ colony forming units/ml (CFU/ml). Resazurin was used for the indication of bacterial growth, as described in the preceding section (Fig 2B).

The potential synergistic effect of two drugs was calculated using the fractional inhibitory concentration index (FICI) according to the following formula [10]:

$$FICI = \frac{MIC_A \text{ in combination}}{MIC_A} + \frac{MIC_B \text{ in combination}}{MIC_B}$$

In most studies where the checkerboard assay is used, the FICI cutoff to define synergy was set to < 0.5. A range between 0.5 and 1.0 was defined as additive, while 1.0 to 4.0 was interpreted as indifferent. Values of > 4.0 were considered antagonistic [27].

## Growth kinetics

To determine the kinetics of bacterial growth, the optical density at 600 nm ($OD_{600}$) was measured continuously for 24 h at 37°C with shaking at 600 rpm (SPECTROstar$^{Nano}$, BMG Labtech). For visualization, blank-corrected data derived from raw measurements were used. Growth kinetics contribute to understanding the functions of microbial cells and their behavior under different conditions, such as in response to antibiotics [28–30].

## Time-kill assay

Time-kill assays were conducted to evaluate the bactericidal kinetics and potential synergistic interactions of FTIs and colistin [31]. Bacterial suspensions were adjusted to 0.5 McFarland and diluted 1:200 in MHB to obtain an initial inoculum of $5 \times 10^5$ CFU/ml. Cultures were incubated at 37°C with shaking at 180 rpm. At defined time points (0, 1, 2, 4, 8, and 24 h), samples were withdrawn, serially diluted in PBS, and 10 µl were spot plated onto MHA. CFU/ml were determined after overnight incubation at 37°C. Bactericidal activity was defined as a $\geq 3 \log_{10}$ CFU/ml reduction from the initial inoculum. Synergy was defined as a $\geq 2 \log_{10}$ CFU/ml reduction compared to the most active single agent [32].

## NPN uptake assay

To assess compound-induced changes in outer membrane permeability, N-Phenyl-1-naphthylamine (NPN; Sigma-Aldrich) uptake assays were performed [33,34]. Briefly, bacteria were cultured to exponential phase ($OD_{600}$ 0.3–0.6), harvested, washed twice, and resuspended in assay buffer (5 mM HEPES, 5 mM glucose, pH 7.2) to $OD_{600}$ 1.0. 100 µl of cell suspension were mixed with 100 µl of assay buffer containing 20 µM NPN in black 96-well plates. Fluorescence (excitation 350 nm, emission 420 nm) was monitored every 30 s over 10 min at room temperature. NPN uptake was calculated using

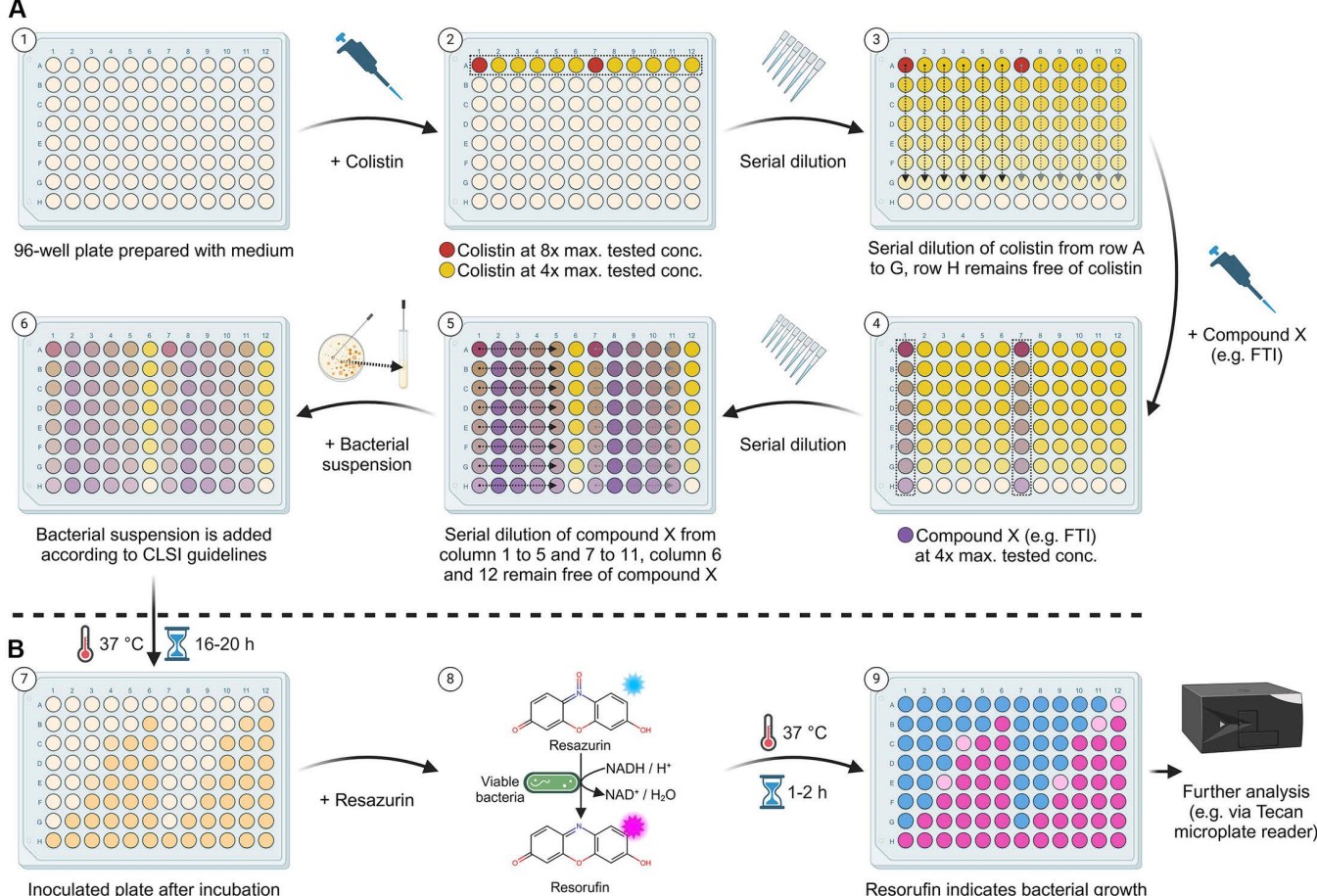

**Fig 2. Overview of the checkerboard assay.** (A) illustrates the setup process: (1) The appropriate medium, typically MHB, is added to all wells of a 96-well plate. (2) Colistin is added and mixed to the first row, ensuring that the duplicate preparation uses either four or eight times the target concentration, indicated by red or yellow dots. (3) A serial dilution is performed from row A to G, leaving row H free of colistin. (4) The test substance, such as an FTI, is added and mixed to columns 1 and 7, using four times the target concentration. (5) Serial dilutions are then conducted from columns 1 to 5 and 7 to 11, leaving columns 6 and 12 free of the test substance. (6) The plate is inoculated following CLSI guidelines. The color gradients represent the varying concentrations of the substances, while arrows indicate the direction of serial dilutions. The use of a single plate for two tests promotes environmental sustainability. (B) depicts the steps post-incubation: (7) Preliminary results can be observed based on visible bacterial growth or absence thereof. (8) For enhanced visualization, resazurin is added, which is reduced by viable cells to fluorescent resorufin, highlighting cell viability. (9) After a short incubation (1–2 h), fluorescence can be measured for more precise analysis. Additional quantitative assessments may be performed using a microplate reader. Created in BioRender.

the following formula (Muheim et al., 2017), where $F_{obs}$ is the observed fluorescence at a given compound concentration, $F_{con}$ is the fluorescence of NPN with cells plus solvent control, and $F_B$ is the fluorescence of NPN in the absence of cells. Data over 10 minutes was averaged.

$$NPN\ uptake = (F_{obs} - F_B) - (F_{con} - F_B)$$

## Statistics

Data processing and statistical analysis were performed using Python 3.12.7 and SciPy 1.14.1 [35]. All data were obtained from at least three independent experiments and presented as means ± standard deviation (SD), unless otherwise noted.

For growth kinetics, cubic spline interpolation was applied for data smoothing. For time-kill evaluation, statistical significance was assessed using one-way ANOVA with Tukey's post hoc test for multiple comparisons. For NPN uptake analysis, one-way ANOVA with Dunnett's post hoc test was used to compare colistin-treated samples to the control. Significance is indicated as follows: $p < 0.05$ (*), $p < 0.01$ (**), $p < 0.001$ (***).

## Results

### Antimicrobial susceptibility

To evaluate the FICI and classify the synergistic potential of drugs in combination with colistin, colistin was tested on all strains to determine its MIC (Table 2). In addition, we indicate clinical breakpoints and the respective epidemiological cut-off values (ECOFF), as far as they are provided by the European Committee on Antimicrobial Susceptibility Testing (EUCAST) [36]. Regular antibiotics were tested simultaneously as controls for antimicrobial susceptibility using resazurin as an indicator of cell viability (Fig 3A). Furthermore, gentamicin and vancomycin were tested on *S. aureus* and methicillin-resistant *S. aureus* (Fig 3B). RFI values below 10% correspond in all cases with the absence of bacterial growth. All controls are listed in the supplementary files where additional technical information is provided (S1 File).

The tested bacterial strains show very diverse sensitivity profiles. In the case of gram-positive bacteria, the difference between *S. aureus* and MRSA is particularly pronounced. MRSA is resistant to the quantities of ampicillin, chloramphenicol, kanamycin and tetracycline selected in this study. The MRSA strain tested is also less sensitive to gentamicin, while there are no noticeable differences between *S. aureus* and MRSA with vancomycin. Among the gram-negative strains tested, *A. baumannii* and *P. paraeruginosa* are almost exclusively susceptible to colistin and not to the four control antibiotics.

### Synergy assessment

When farnesyltransferase inhibitors were combined with sub-inhibitory concentrations of colistin, ranging from 0.008 to 0.5 µg/ml for all bacteria except *E. cloacae*, where concentrations ranged from 0.06 to 4.0 µg/ml, the growth of gram-positive

**Table 2. MIC values for colistin determined for the tested bacterial strains.**

| Bacterial strain | | Colistin | | |
|---|---|---|---|---|
| | | MIC [µg/ml] | Clinical breakpoints | ECOFF [µg/ml] |
| gram-positive | *B. subtilis* | 16.0 | N/A[b] | N/A |
| | *E. faecium* | No inhibition[a] | N/A | N/A |
| | *S. aureus* | No inhibition | N/A | N/A |
| | *S. aureus* (MRSA) | No inhibition | N/A | N/A |
| | *S. epidermidis* | No inhibition | N/A | N/A |
| gram-negative | *A. baumannii* | 1.0 | Susceptible (≤ 2.0) | 2.0 |
| | *E. cloacae* | 32.0[c] | Resistant (> 2.0) | 2.0 |
| | *E. coli* | 0.5 | Susceptible (≤ 2.0) | 2.0 |
| | *K. pneumoniae* | 1.0 | Susceptible (≤ 2.0) | 2.0 |
| | *P. paraeruginosa* | 1.0 | Susceptible (≤ 4.0) | 4.0[d] |

N/A indicates data not available.

[a]No inhibitory effects were observed at the highest tested colistin concentration for *E. faecium*, *S. aureus*, methicillin-resistant *S. aureus* (MRSA), and *S. epidermidis*.

[b]EUCAST does not specify clinical breakpoints or ECOFF values for colistin on gram-positive bacteria.

[c]*E. cloacae* exhibited a pronounced tendency toward colistin heteroresistance, complicating the determination of a specific MIC.

[d]For *P. paraeruginosa*, the ECOFF value for *P. aeruginosa* was applied.

**A**

| | Ampicillin [µg/ml] | | | Chloramphenicol [µg/ml] | | | Kanamycin [µg/ml] | | | Tetracycline [µg/ml] | | |
|---|---|---|---|---|---|---|---|---|---|---|---|---|
| | 8.0 | 4.0 | 2.0 | 32.0 | 16.0 | 8.0 | 8.0 | 4.0 | 2.0 | 8.0 | 4.0 | 2.0 |
| *B. subtilis* (gram-positive) | | | | | | | | | | | | |
| *E. faecium* | | | | | | | | | | | | |
| *S. aureus* | | | | | | | | | | | | |
| *S. aureus* (MRSA) | | | | | | | | | | | | |
| *S. epidermidis* | | | | | | | | | | | | |
| *A. baumannii* (gram-negative) | | | | | | | | | | | | |
| *E. cloacae* | | | | | | | | | | | | |
| *E. coli* | | | | | | | | | | | | |
| *K. pneumoniae* | | | | | | | | | | | | |
| *P. paraeruginosa* | | | | | | | | | | | | |

RFI [%]: < 5.0 | 5.0 - 10.0 | 10.0 - 15.0 | 15.0 - 20.0 | 20.0 - 25.0 | 25.0 - 30.0 | 30.0 - 50.0 | > 50.0

**B**

| | Gentamicin [µg/ml] | | | | | | | | | | | |
|---|---|---|---|---|---|---|---|---|---|---|---|---|
| | 16.0 | 8.0 | 4.0 | 2.0 | 1.0 | 0.5 | 0.25 | 0.125 | 0.06 | 0.032 | 0.016 | 0 |
| *S. aureus* | | | | | | | | | | | | |
| *S. aureus* (MRSA) | | | | | | | | | | | | |

| | Vancomycin [µg/ml] | | | | | | | | | | | |
|---|---|---|---|---|---|---|---|---|---|---|---|---|
| | 16.0 | 8.0 | 4.0 | 2.0 | 1.0 | 0.5 | 0.25 | 0.125 | 0.06 | 0.032 | 0.016 | 0 |
| *S. aureus* | | | | | | | | | | | | |
| *S. aureus* (MRSA) | | | | | | | | | | | | |

**Fig 3. Relative fluorescence intensity (RFI) values for the tested bacterial strains and antibiotics.** (A) presents the RFI values for gram-positive and gram-negative bacterial strains treated with varying concentrations of ampicillin, chloramphenicol, kanamycin, and tetracycline. (B) focuses on the RFI values for *S. aureus* and MRSA treated with gentamicin and vancomycin. A high RFI value indicates active bacterial metabolism and growth, as resazurin is reduced to fluorescent resorufin by viable cells. Conservatively, an RFI below 10% is consistently correlated with the absence of bacterial growth, enabling the assessment of antimicrobial effects and determination of MIC values.

and gram-negative bacteria was inhibited. The effects of FTIs and colistin were very different in the ten bacterial strains tested. The complete data can be found in the supplementary files (S1–S10 Figs).

Among the gram-negative strains, the non-peptidomimetic FTIs lonafarnib and tipifarnib notably reduced the MIC of colistin. For instance, in *A. baumannii*, the colistin MIC decreased from 1.0 µg/ml to 0.25 µg/ml (four-fold reduction) when 15.6 µM lonafarnib was applied. This effect was even more pronounced with tipifarnib; at 31.3 µM, the colistin MIC was reduced to 0.125 µg/ml (eight-fold reduction). These effects were comparable for *E. coli* and *K. pneumoniae*, though the *K. pneumoniae* strain exhibited a tendency to form subcolonies, particularly noticeable with lonafarnib treatment. Despite this, strong synergy (FICI < 0.5) was observed in some cases. In contrast, *P. paraeruginosa* largely lacked synergistic interactions, with only tipifarnib showing a slight additive effect. Lonafarnib, even at high concentrations combined with half the colistin MIC, exhibited minimal inhibitory activity against this strain. Conversely, in *E. cloacae*, lonafarnib displayed stronger synergistic effects than tipifarnib, reducing the colistin MIC from 32 µg/ml to 1.0 µg/ml (32-fold reduction) when 15.6 µM lonafarnib was used.

In comparison, the peptidomimetic FTIs, B581 and FTI-277, exhibited weaker effects on gram-negative bacteria. Additive and partial synergistic effects were observed but were less pronounced than those seen with lonafarnib and tipifarnib. For *E. coli*, FTI-277 reduced the colistin MIC to 0.125 µg/ml when 31.3 µM was applied. However, in *K. pneumoniae*, subcolonies reappeared, particularly with FTI-277, leading to inconsistent growth inhibition across replicates and complicating the evaluation of synergy. Similarly, both B581 and FTI-277 showed negligible effects on *P. paraeruginosa* and *E. cloacae*.

Finally, αHFP and bempedoic acid exhibited no activity against the gram-negative bacteria tested. Fig 4 summarizes the results for the gram-negative strains, illustrating the investigated properties in relation to the inhibitory effects and identified synergistic activities.

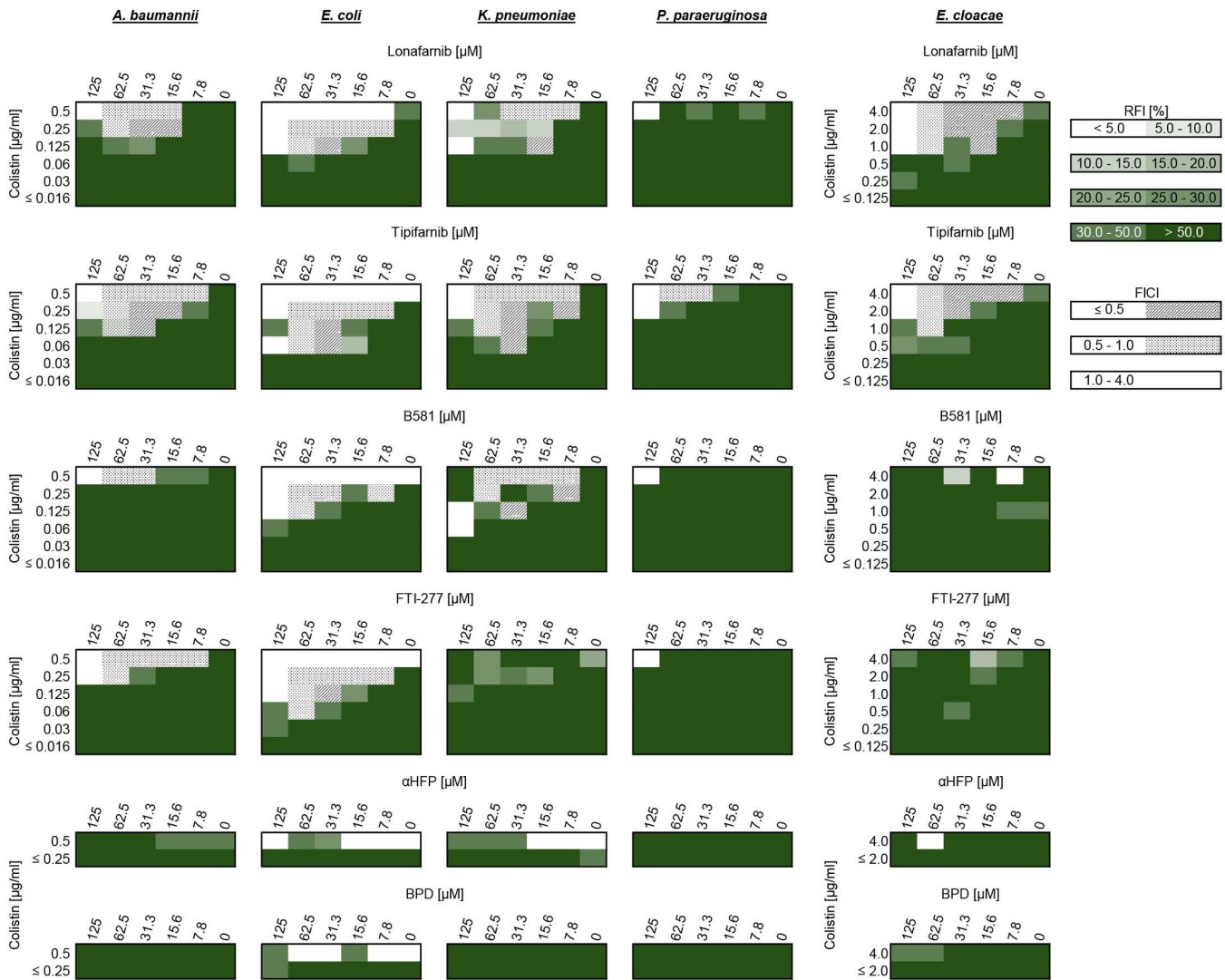

**Fig 4. Relative fluorescence intensity (RFI) values for the tested gram-negative bacterial strains and substances, along with corresponding fractional inhibitory concentration index (FICI) values.** The bacterial strains are arranged alphabetically from left to right, with *E. cloacae* positioned separately on the far right due to the use of different colistin concentrations for this strain. FICI values are visualized to classify interactions: values below 0.5 indicate strong synergy, values between 0.5 and 1.0 represent additive effects, and values from 1.0 to 4.0 suggest indifference. No antagonistic interactions (FICI > 4.0) were observed. The RFI values for αHFP and bempedoic acid (BPD) are only partially displayed, as these substances demonstrated no inhibitory effects.

The tested gram-positive bacteria differ significantly from the gram-negative strains. No effects of B581, FTI-277, αHFP, or BPD were observed in any of the gram-positive representatives. However, lonafarnib and tipifarnib exhibited strong inhibitory effects on MRSA, *S. aureus*, and *S. epidermidis*, independent of colistin. Thus, no synergistic effects were identified in these cases and colistin did not compromise the inhibitory effects of lonafarnib or tipifarnib on these bacterial strains.

MRSA was inhibited at a lonafarnib concentration of 31.3 µM, whereas non-methicillin-resistant *S. aureus* showed inhibition at 15.6 µM. Similar MIC values were observed for tipifarnib against these strains; however, *S. aureus* exhibited inconsistent behavior with tipifarnib. At lower concentrations, tipifarnib partially reduced growth without achieving complete

inhibition. Both lonafarnib and tipifarnib affected *S. epidermidis* similarly. Conversely, *E. faecium* showed no growth inhibition with either lonafarnib or tipifarnib when standard microdilution methods were applied. It is important to note that the *E. faecium* strain used in this study displayed an overall slow growth rate, which was further reflected in its generally low metabolic activity, as indicated by a slower reduction of resazurin.

A unique case among the gram-positive bacteria was *B. subtilis*. This was the only gram-positive organism successfully inhibited by colistin (16 µg/mL) in this study. Furthermore, lonafarnib exhibited inhibitory effects on *B. subtilis* growth only in combination with colistin. A successful inhibition was observed with 31.3 µM lonafarnib and 0.5 µg/mL colistin, representing a 32-fold reduction in the required colistin concentration. Since colistin alone demonstrated activity against *B. subtilis*, an additive effect with tipifarnib and colistin is inferred. At a tipifarnib concentration of 15.6 µM, the colistin exhibited measurable effects. These findings are summarized in Fig 5.

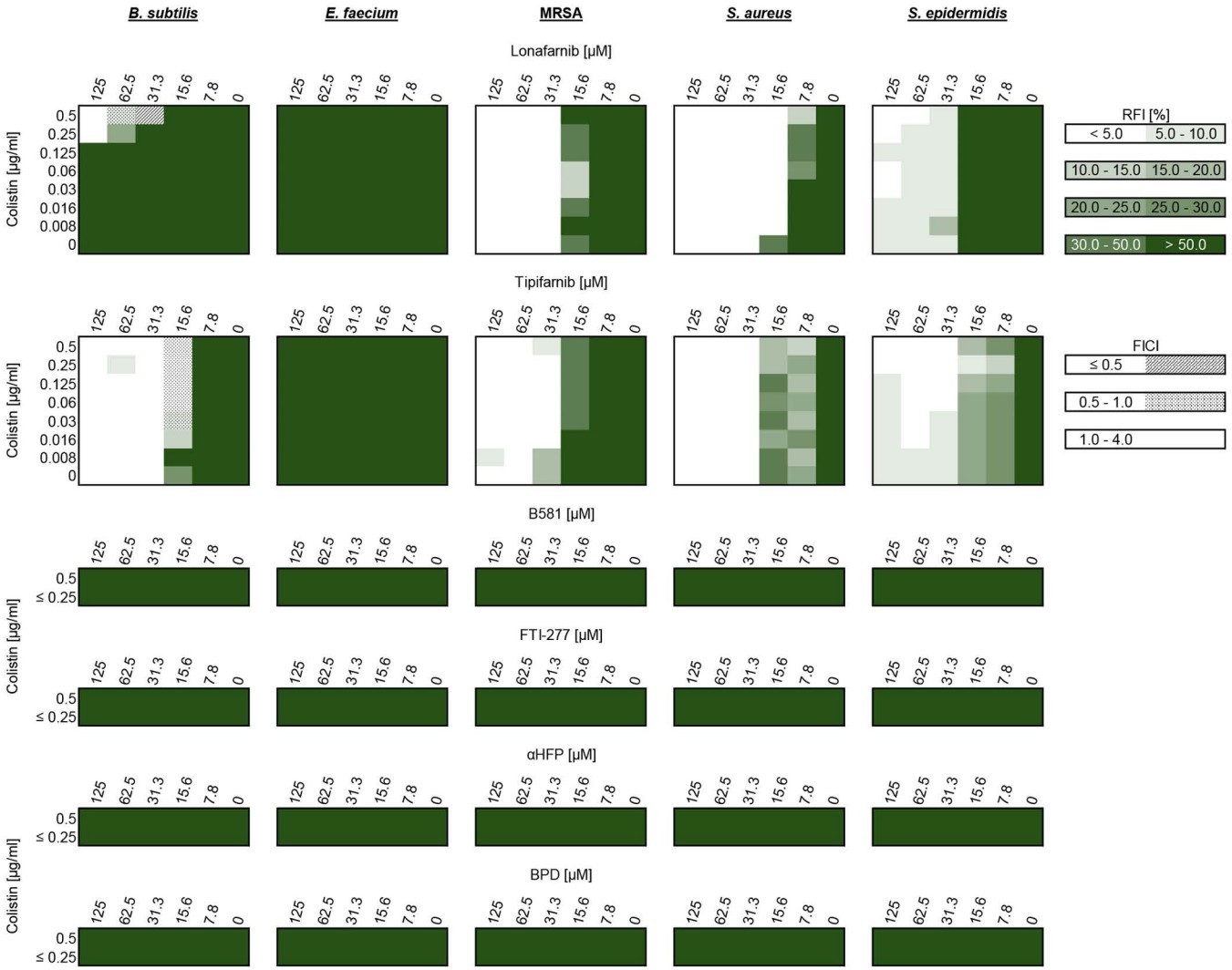

**Fig 5. Relative fluorescence intensity (RFI) values for the tested gram-positive bacterial strains and substances, along with fractional inhibitory concentration index (FICI) values.** The bacterial strains are arranged alphabetically from left to right. FICI values are visualized for *B. subtilis*, with no antagonistic effects observed (FICI > 4.0). The RFI values for B581, FTI-277, αHFP, and bempedoic acid (BPD) are only partially displayed, as these substances showed no inhibitory effects.

To summarize the findings across all bacterial strains, the overall highest FICI values and fold changes in colistin MIC are presented as heatmaps in Fig 6. These data illustrate strain-specific variations in FTI/colistin interactions and colistin MIC reductions.

Lonafarnib and tipifarnib were the only FTIs that exhibited effects against gram-positive bacteria. However, synergistic effects (FICI < 0.5) were observed exclusively in *B. subtilis*, while *E. faecium* remained unaffected by any of the tested substances under the given conditions. In contrast, all tested FTIs inhibited gram-negative bacteria, with lonafarnib and tipifarnib demonstrating the highest FICI values in combination with colistin, particularly against *E. cloacae* and *K. pneumoniae* (FICI < 0.25). *P. paraeruginosa* was the least affected among the gram-negative strains (Fig 6A).

Within the gram-positive bacterial strains tested in this study, only *B. subtilis* exhibited a change in the required colistin MIC. A 32-fold reduction in the colistin concentration was observed when lonafarnib was administered concurrently. Although a certain additive effect was also observed with the simultaneous administration of colistin and tipifarnib, no fold change (FC) in colistin MIC was reported. This is because tipifarnib, at a concentration of 31.3 µM, inhibited the growth of *B. subtilis* independently of colistin. For gram-negative bacteria, FICI values strongly correlated with colistin MIC FCs. The most pronounced effect was observed in *E. cloacae*, where lonafarnib and tipifarnib both reduced the required colistin concentration from 32.0 µg/ml to 1.0 µg/ml, corresponding to a 32-fold reduction in colistin MIC in a concentration-dependent manner. In contrast, peptidomimetic FTIs failed to reduce the colistin MIC in *E. cloacae*. This was not the case for other gram-negative strains, where reductions were observed. Yet, *K. pneumoniae* was affected by B581 but not by FTI-277 (Fig 6B).

## Growth kinetics and time-kill

While the checkerboard assays explored the synergistic effects of FTIs and colistin across both gram-positive and gram-negative bacteria, growth kinetics experiments were conducted to gain deeper insights into the temporal dynamics of FTI activity, focusing specifically on *S. aureus*, a clinically significant gram-positive pathogen, and *E cloacae*, a colistin resistant strain. These experiments aimed to complement the endpoint-based MIC and FICI results by evaluating

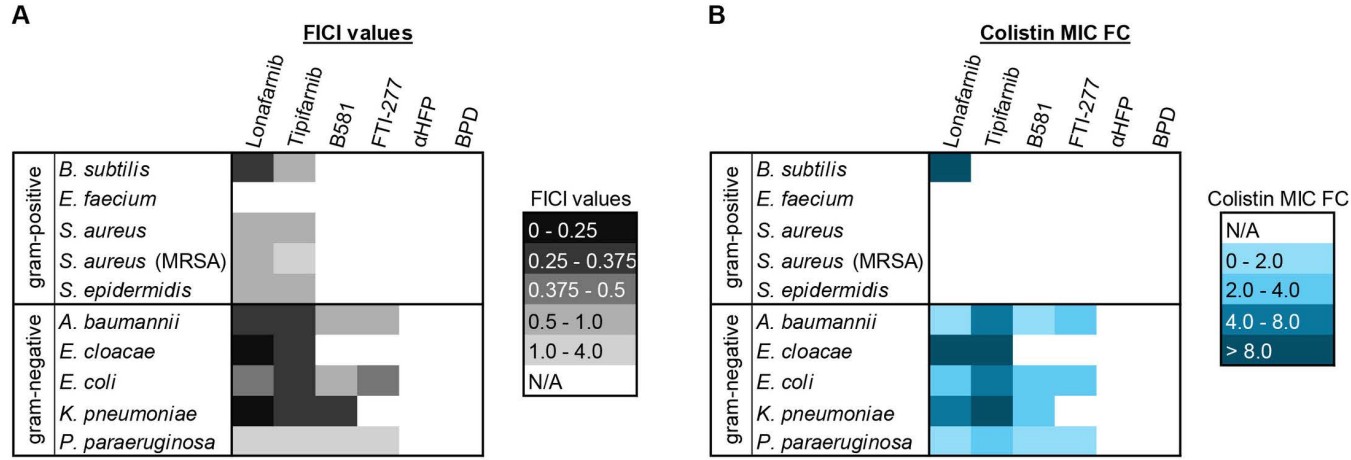

**Fig 6. Heatmap comparison of the highest fractional inhibitory concentration index (FICI) values and fold change (FC) reductions in colistin MIC across all tested bacterial strains.** (A) depicts strain-specific synergy between FTIs and colistin, with lonafarnib and tipifarnib exhibiting the strongest effects against gram-negative bacteria such as *E. cloacae* and *K. pneumoniae*. (B) highlights fold changes in colistin MIC, showcasing substantial reductions, including a 32-fold decrease in *B. subtilis* (with lonafarnib) and *E. cloacae* (with both lonafarnib and tipifarnib). These heatmaps provide an overview of differential bacterial responses, emphasizing the potential of FTI/colistin combinations to enhance antimicrobial efficacy and reduce colistin requirements.

the consistency and duration of growth inhibition under dynamic conditions. To this end, *S. aureus* was tested with four concentrations of lonafarnib and tipifarnib, respectively, using continuous $OD_{600}$ measurements over 24 h at 37°C with shaking at 600 rpm. These tests revealed slight deviations compared to standard CLSI-based methods. At a concentration of 7.8 µM lonafarnib, the growth of *S. aureus* was completely inhibited for the first 16 h, after which rapid bacterial growth resumed (Fig 7A). Higher concentrations of 15.6 µM and 31.3 µM lonafarnib resulted in continuous growth inhibition throughout the 24-hour measurement period. A similar pattern was observed for tipifarnib: at 7.8 µM, minimal growth was detectable as a slight increase in $OD_{600}$ after 12 h (Fig 7C). Higher concentrations also led to sustained inhibition of growth. The observed differences may indicate an influence of shaking on the experimental outcomes, particularly regarding compound distribution in the medium.

Growth kinetics were similarly assessed for *E cloacae*. Unlike *S. aureus*, growth inhibition was not observed with lonafarnib or tipifarnib alone at concentrations up to 31.3 µM, or with sub-inhibitory concentrations of colistin. However, clear growth suppression was observed in combination treatments with 4 µg/ml of colistin and 15.6 µM or 31.3 µM of lonafarnib, which maintained growth inhibition for the entire 24-hour measurement period. Even a concentration of 7.8 µM of lonafarnib combined with colistin resulted in a transient growth delay compared to colistin alone (Fig 7E). Tipifarnib/sub-inhibitory colistin had a similar effect on growth kinetics, albeit less pronounced. Growth was prevented over 24 h at 31.3 µM tipifarnib plus 4 µg/ml colistin, while combinations at 15.6 µM caused a delayed onset of growth by about 10 h (Fig 7G).

Time-kill assays were conducted to gain more insights into the bactericidal kinetics of FTIs and colistin. Due to *S. aureus's* inherently slower growth rate compared to *E. cloacae*, measurable differences in bacterial counts emerged after 4 h. By 8 h, all tested concentrations of lonafarnib had resulted in a significant reduction of over 3 $\log_{10}$ CFU/ml compared to the DMSO control (p < 0.001) (Fig 7B). After 24 h, this reduction remained significant across all concentrations, although the effect of 7.8 µM was notably less pronounced. A similar effect was observed for tipifarnib: while all tested concentrations demonstrated bactericidal activity, the highest concentration (31.3 µM) led to a reduction of approximately 7 $\log_{10}$ CFU/ml compared to the control (p < 0.0001). There was a statistically significant difference between the 31.3 µM and 15.6 µM concentrations of tipifarnib (p < 0.0001), indicating a concentration-dependent effect (Fig 7D).

For *E. cloacae*, the results for lonafarnib (Fig 7F) and tipifarnib (Fig. 7H) were highly comparable. Neither compound exhibited any bactericidal activity when used alone, with no reduction in CFU/ml relative to the DMSO control. Colistin, when applied at sub-inhibitory concentrations, induced a transient reduction in bacterial counts, most notably at the earliest time points. After 4 h, treatment with colistin alone reduced bacterial load by approximately 5 $\log_{10}$ CFU/ml compared to the control (p < 0.001). However, this effect was not sustained; by 8 h, the reduction no longer reached the ≥ 2 $\log_{10}$ CFU/ml threshold for bactericidal activity, and by 24 h, CFU levels had returned to control levels. In contrast, the combined approach demonstrated clear synergy. When combined with sub-inhibitory concentrations of colistin, both lonafarnib and tipifarnib resulted in a ≥ 3 $\log_{10}$ CFU/ml reduction compared to colistin alone, fulfilling the criterion for synergistic bactericidal activity. These differences were highly significant at the 2-hour time point for both combinations (p < 0.0001).

## Outer membrane permeability

Focusing on the outer membrane of gram-negative bacteria, permeability was assessed for *E. cloacae* and *E. coli* using the fluorescent dye NPN. This small molecule exhibits low fluorescence in an aqueous solution but becomes strongly fluorescent when it binds to phospholipids. Under normal conditions, uptake is limited and fluorescence remains low. However, when the OM is compromised, NPN can penetrate more easily. This results in increased fluorescence due to its interaction with membrane lipids [33]. The data were normalized to the highest tested colistin concentration (2 × MIC).

In *E. cloacae*, all tested colistin concentrations – including sub-inhibitory levels down to 1/8 MIC – led to a highly significant increase in NPN uptake compared to the DMSO control (p < 0.0001 for all conditions). While the increased fluorescence at 1/2 MIC and 1/4 MIC could be partly explained by cell lysis, this is unlikely at 1/8 MIC, despite the NPN signal remaining significantly elevated (Figs 8A–B). For *E. coli*, all tested concentrations – including sub-inhibitory levels

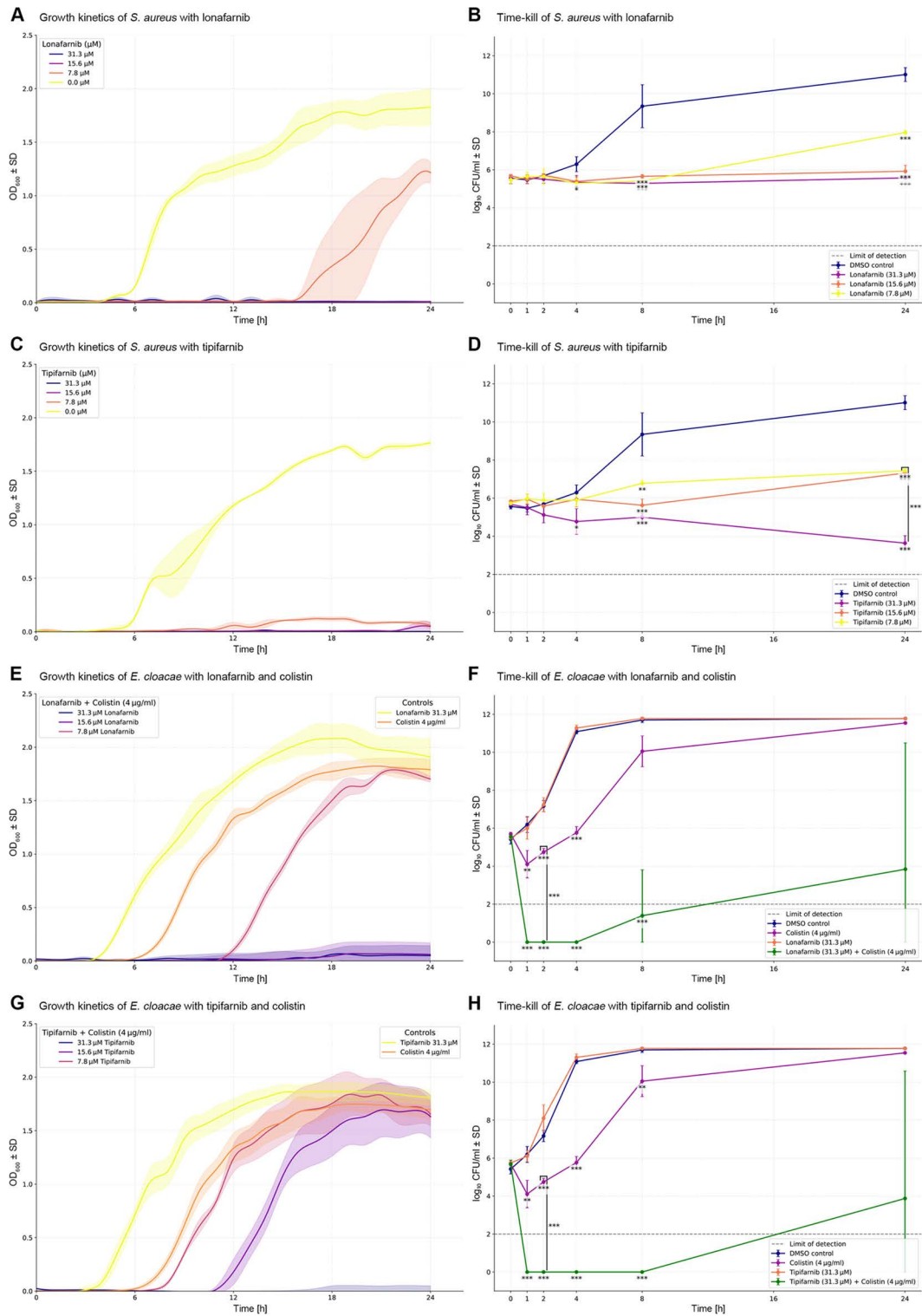

**Fig 7. Growth kinetics and time-kill assays of methicillin-sensitive *Staphylococcus aureus* and colistin-resistant *Enterobacter cloacae*.** Growth kinetics are shown as the mean optical density (OD) at 600 nm plotted against different substance concentrations or combinations, with each concentration of combination represented by a different color. Standard deviation (SD) values are indicated as shaded areas around the primary plot lines. Cubic spline interpolation was applied to smooth the data. Panels (A) and (C) show the growth kinetics for *S. aureus*, while panels (E) and (G) display

the results for *E. cloacae*. To assess bactericidal activity over time, time-kill assays were performed. Panels (B) and (D) show the time-kill results for *S. aureus*, and panels (F) and (H) for *E. cloacae*. Statistical analysis was conducted using one-way ANOVA with Tukey's post hoc test for multiple comparisons. Significance is indicated as follows: $p < 0.05$ (*), $p < 0.01$ (**), $p < 0.001$ (***). A dashed line at $\log_{10} 2$ indicates the limit of detection, corresponding to 100 CFU/ml. Values below this threshold could not be reliably quantified due to technical reasons. For the time-kill assays, lonafarnib and tipifarnib were assessed simultaneously. Visualization was split to allow better distinction. This resulted in shared controls for panels (B) and (D), and for panels (F) and (H).

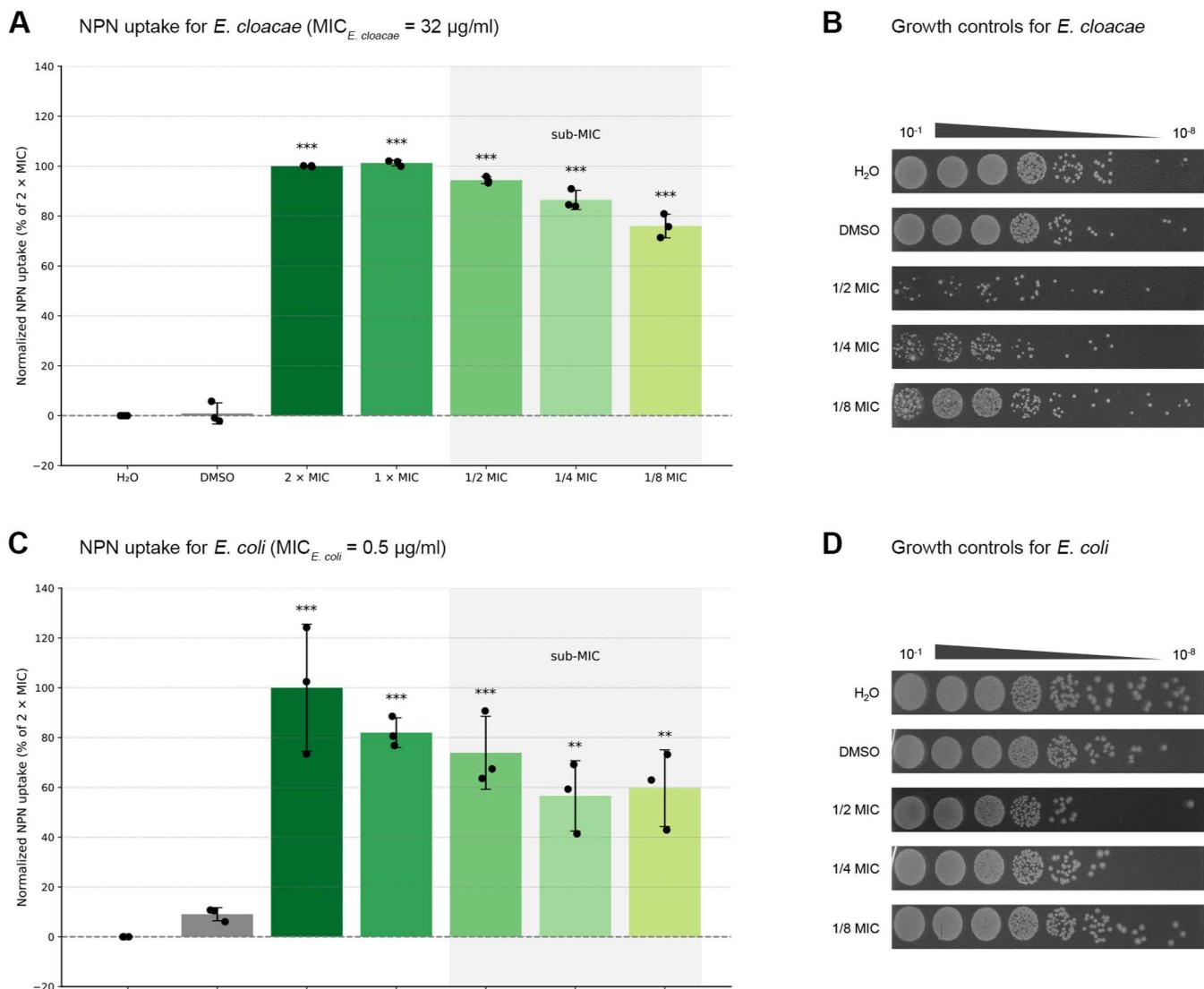

**Fig 8. Sub-inhibitory concentrations of colistin increase outer membrane permeability in *E. cloacae* (A) and *E. coli* (C).** The integrity of the outer membrane (OM) was assessed using the hydrophobic fluorescent probe N-phenyl-1-naphthylamine (NPN), which does not invade cells with an intact OM, but fluoresces when inserted into the phospholipid bilayer upon OM compromise. The effect of different concentrations of colistin on OM permeability was determined. Data were normalized to the 2×MIC colistin condition. DMSO served as a negative control and $H_2O$ as the colistin solvent. Statistical analysis was conducted using one-way ANOVA with Dunnett's post hoc test for multiple comparisons. Significance is indicated as follows: $p < 0.05$ (*), $p < 0.01$ (**), $p < 0.001$ (***). The sub-inhibitory concentration range is indicated by a light grey box ("sub-MIC"). To control for antimicrobial activity, dilution series were spotted on MHA to confirm that sub-inhibitory concentrations of colistin alone do not inhibit the growth of *E. cloacae* (B) and *E. coli* (D).

– resulted in a statistically significant increase in NPN uptake compared to the DMSO control ($p < 0.0001$ to $p = 0.0087$) (Figs 8C–D). These findings indicate that even sub-inhibitory concentrations of colistin can increase outer membrane permeability in both species.

## Discussion

In this study, the effects of different farnesyltransferase inhibitors in combination with colistin were investigated to identify potential synergistic effects and expand the spectrum of FTIs to gram-negative bacteria. The main purpose of FTIs is to inhibit FTase, an enzyme that, in eukaryotic cells, catalyzes the post-translational transfer of a $C_{15}$ (farnesyl) isoprenoid moiety from farnesyl pyrophosphate (FPP) to a cysteine residue in target protein [37]. Together with geranylgeranyltransferases, FTase plays a central role in protein prenylation [38,39].

The formation of isoprenoids, necessary for the prenylation, such as FPP and geranylgeranyl pyrophosphate (GGPP), relies on either the mevalonate pathway or, more commonly in bacteria, the methylerythritol phosphate (MEP) pathway [40,41]. Independently of the biosynthetic pathway used, isoprenoids are universally distributed across all domains of life – bacteria, archaea, and eukaryotes – and are essential for cell survival, including in bacterial pathogens [42]. This universality underscores their critical role in various cellular processes. Interestingly, the antimicrobial effects of FTIs appear to be independent of the biosynthetic pathway for isoprenoid production, as FTIs effectively inhibit the growth of bacteria utilizing both the MEP and mevalonate pathways [43]. Isoprenoids are vital for bacterial cells, serving roles in pathogenicity [44,45] and acting as intermediates in cell wall biosynthesis [46]. However, their utilization in protein prenylation within bacterial cells remains largely elusive [5].

As a ubiquitous covalent post-translational modification in eukaryotic cells, protein prenylation involves the attachment of isoprenoids like FPP or GGPP to proteins [47]. In contrast, bacteria rarely employ such modifications, as post-translational modifications are generally less prevalent in prokaryotes [48]. The extent of bacterial protein modification varies significantly among species and is strongly influenced by environmental conditions. Nevertheless, post-translational modifications can profoundly impact bacterial physiology and virulence regulation, altering protein structure, activity, localization, and biomolecular interactions [49]. While no evidence suggests that bacteria utilize an inherent protein prenylation machinery for post-translational modifications, some pathogenic bacteria exploit host-cell prenylation systems. For instance, bacterial effector proteins with a C-terminal CAAX motif have been shown to be prenylated by the host's prenylation machinery. This phenomenon has been observed in pathogens such as *Salmonella typhimurium* and *Legionella pneumophila* [50–52]. Additionally, several other bacterial proteins, primarily from pathogenic strains, have been identified as potential targets for host-mediated prenylation, including possible farnesylation [53]. While endogenous bacterial protein prenylation remains largely unexplored, some exceptions exist. For instance, *B. subtilis* utilizes a distinct prenylation system for the quorum-sensing protein ComX, where a farnesyl group is transferred to a tryptophan residue by ComQ, rather than by FTase [54]. Notably, the antimicrobial effects observed in our study are also independent of host prenylation systems, suggesting that bacterial prenylation-related mechanisms may be relevant targets. Nonetheless, the reliance of these pathogens on host prenyltransferases does not preclude the possibility that bacterial prenyltransferases may also exist [5]. Therefore, the development of FTIs, which is already conducted in the field of cancer therapy, could also be of great importance in the treatment of infectious diseases of various kinds. For instance, lonafarnib is under clinical investigation for the treatment of hepatitis D virus (HDV) infections. Additionally, lonafarnib shows promising effects for managing bronchiolitis and pneumonia caused by respiratory syncytial virus (RSV) and has also demonstrated effects against SARS-CoV-2 [55–57]. Recent studies have also highlighted the antimicrobial effects of FTIs. We demonstrated significant antimicrobial activity of lonafarnib and tipifarnib – two non-peptidomimetic FTIs – against gram-positive bacteria, including *S. aureus*, *S. epidermidis*, and even methicillin-resistant *S. aureus*. However, no activity of these compounds against gram-negative bacteria was reported, such as *E. coli*, *K. pneumoniae*, and *P. aeruginosa*. It was hypothesized that the lack of activity was due to the OM of gram-negative bacteria, which serves as an additional intrinsic permeability barrier that is absent in gram-positive bacteria [5].

In the present study, colistin, a well-established antimicrobial from the polymyxin class, was used to overcome this limitation in order to use the FTIs to treat a broad range of bacterial strains. Colistin disrupts the OM of gram-negative bacteria by binding to lipopolysaccharides (LPS) via electrostatic interactions, ultimately leading to bacterial cell death [8,58]. When sub-inhibitory concentrations of colistin were applied to compromise the OM, lonafarnib and tipifarnib exhibited significant antimicrobial effects against gram-negative bacteria. Additionally, this approach enabled the peptidomimetic FTIs, B581 and FTI-277, to inhibit the growth of gram-negative bacterial strains. Interestingly, these two peptidomimetic FTIs did not show any activity against gram-positive bacteria, in contrast to lonafarnib and tipifarnib. This difference is likely attributable to their distinct chemical properties, despite sharing the same enzymatic target in eukaryotic cells. Notably, lonafarnib and tipifarnib are halogenated, whereas the peptidomimetic FTIs are not. Halogenated compounds are known to exhibit increased affinity for gram-positive bacteria [59], which may explain the observed activity patterns, and should be further studied. The strategy of using sub-inhibitory concentrations of colistin is well-established in the field of drug repurposing to evaluate the antimicrobial activity of compounds against gram-negative bacteria. Statins, for example, which are widely used as lipid-modifying agents to control hyperlipidemia, have demonstrated antimicrobial properties [4]. This unexpected effect was initially observed as a pleiotropic property of these drugs. Epidemiological studies further suggested a correlation between statin therapy and a reduced risk of sepsis and other severe bacterial infections [60,61]. Initial susceptibility testing revealed that statins exhibit antimicrobial activity, though the clinical relevance remains uncertain. This is because the *in vivo* concentrations achieved with clinical statin doses are significantly lower than those required for antimicrobial activity [62]. Nonetheless, subsequent studies have highlighted the potential of statins as novel antimicrobials and demonstrated their synergistic effects even on gram-negative bacteria when combined with colistin [63,64]. Although the mechanism of action (MOA) of statins in bacteria is still not fully elucidated, in humans, statins exert their primary effect by inhibiting class I HMG-CoA reductase within the mevalonate pathway [65]. However, only a few bacterial species utilize the mevalonate pathway to produce isopentenyl pyrophosphate and dimethylallyl pyrophosphate [66,67]. Consequently, Thangamani, Mohammad [64] hypothesized that the MOA of simvastatin, a representative statin with notable antimicrobial activity, differs in bacteria due to the absence of the class I HMG-CoA reductase enzyme. Instead, its MOA appears to be complex, involving inhibition of multiple biosynthetic pathways and cellular processes, including selective interference with bacterial protein synthesis [64]. This complexity may be partially adaptable to FTIs, as statins target an earlier step in the mevalonate pathway, upstream of the isoprenoid synthesis required for processes like prenylation. Even further upstream of the mevalonate pathway, another inhibitor that was used in this study, bempedoic acid, exerts its effect. However, bempedoic acid showed no measurable effect on the growth of the bacterial strains tested in this study, regardless of the presence of sub-inhibitory colistin. Bempedoic acid is known to lower low-density lipoprotein cholesterol levels in humans by inhibiting ATP citrate lyase, a key enzyme in the cholesterol biosynthesis pathway that acts upstream of HMG-CoA reductase, the target of statins [19]. While ATP citrate lyase plays a crucial role in autotrophic organisms, such as green sulfur bacteria, by facilitating carbon dioxide fixation [68,69], the gram-positive and gram-negative bacteria tested in this study do not rely on this enzyme for survival.

Furthermore, another inhibitor tested in this study, αHFP, a nonhydrolyzable analog of FPP, functions as a competitive inhibitor of FTase in eukaryotic cells and effectively blocks Ras farnesylation [70,71]. This compound was included in the study to provide a comparative context for the activity of FTIs, as αHFP mimics FPP, the natural substrate of FTase. While αHFP has demonstrated significant effects in eukaryotic systems [72], none of the bacterial strains in this study responded to its presence. This observation highlights the distinct mechanisms of action of FTIs in eukaryotic versus bacterial cells and emphasizes the need for further investigation into their activity in different biological systems.

Further investigation into the effects of FTIs on *B. subtilis*, an important model organism, could be pivotal in elucidating their MOA, particularly regarding interactions with the bacterial cell membrane and wall. This study found that *B. subtilis* strain 168 is not only susceptible to colistin at relatively high concentrations but is also the only gram-positive bacterium among those tested to exhibit a unique inhibition pattern for drug combinations. Colistin likely acts on *B. subtilis* by

stimulating both the TCA cycle and the respiratory chain, enhancing NADH metabolism and leading to oxidative damage, a mechanism that could potentially be applicable to other gram-positive bacteria [73]. While lonafarnib demonstrated a strong synergistic effect only in combination with colistin, tipifarnib appeared to act mostly independently. Notably, this specific strain has been extensively used in research for many years [74]. Recently, Leeten, Jacques [75] highlighted the pleiotropic antimicrobial effects of the antiplatelet drug ticagrelor, employing *B. subtilis* strain 168 in assays specifically designed to study cell wall and membrane alterations.

Although bacteria seem to lack their own prenylation machinery, these findings suggest that host-pathogen interactions involving protein prenylation play a significant role in bacterial virulence. This highlights the broader importance of understanding prenylation in bacterial biology and its potential as a therapeutic target [76]. Building on this concept, the tested FTIs – non-peptidomimetic lonafarnib and tipifarnib, as well as peptidomimetic B581 and FTI-277 – demonstrated the ability to inhibit bacterial growth in this study. While the exact MOA of FTIs in bacteria remains unclear, the observed activity supports the hypothesis that their effects differ from their MOA in eukaryotic cells, potentially involving a complex mechanism with multiple targets that disrupt various biosynthetic pathways and cellular processes. Importantly, these findings demonstrate that FTIs, when combined with sub-inhibitory concentrations of colistin, can extend their antimicrobial activity to gram-negative bacteria. Fig 9 illustrates the current understanding of this process: colistin disrupts the OM of gram-negative bacteria, enabling FTIs to penetrate and exert a concentration-dependent effect on bacterial cells.

FTIs not only demonstrate the ability to inhibit gram-positive and gram-negative bacteria but also significantly reduce the MIC of colistin in these strains. This includes the *E. cloacae* strain used in this study, a member of the C-XI cluster with intrinsic colistin resistance and an associated heteroresistance phenotype [77,78]. MIC values for colistin in this strain vary widely, with this study estimating a value that falls within the expected range for colistin-resistant strains (Table 1). It is important to note that, even under optimal conditions, MIC values should be interpreted as approximate ranges rather

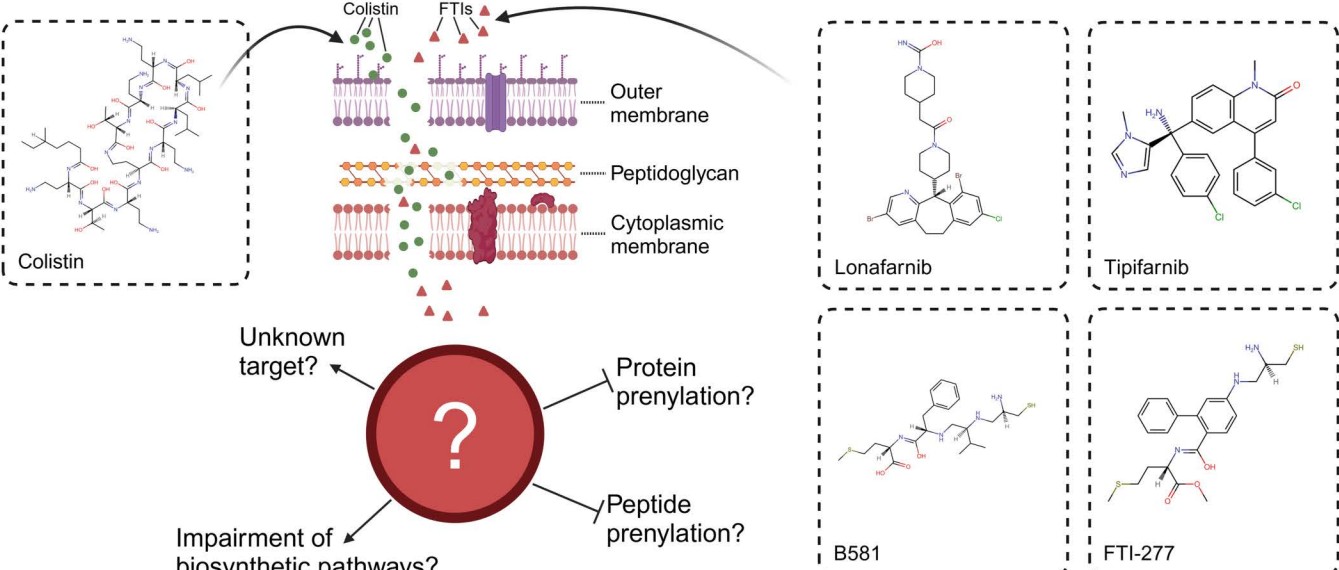

**Fig 9. Proposed mechanism of action for farnesyltransferase inhibitors (FTIs) in combination with colistin on bacteria.** At sub-inhibitory concentrations, colistin disrupts the outer membrane (OM) of gram-negative bacteria by binding to Lipid A in the lipopolysaccharide, compromising the integrity of the OM. This disruption allows FTIs to access their target(s), which remain unidentified. Potential mechanisms include interference with multiple biosynthetic pathways, or disruption of prenylation processes within the bacterial cell. These combined effects ultimately result in bacterial cell death. Created in BioRender.

than precise figures, as variability in measurement is inherent to the method (EUCAST v. 14). However, the strong synergistic effect observed with the combined use of colistin and FTIs supports the hypothesis that FTIs may provide insights into overcoming the growing resistance to colistin. As a member of the ESKAPE pathogen group, *E. cloacae* is responsible for numerous opportunistic nosocomial infections and, less frequently, community-acquired infections. These include urinary tract infections, bacteremia, lower respiratory tract infections, and surgical site infections, as well as colonization of intravascular devices [79]. The *E. cloacae* strain used in this study, which is also the type strain of *E. cloacae* subsp. *cloacae*, exhibits high levels of colistin resistance and is resistant to ampicillin (Fig 3). This strain also demonstrates significant pathogenic potential due to several mechanisms and can, for example, harbor elements such as a type VI secretion system, which is crucial for survival within the host [80,81]. In this study, lonafarnib and tipifarnib reduced the colistin MIC in *E. cloacae* by more than 32-fold (Fig 6). Furthermore, the highest FICI values for synergy were calculated for *E. cloacae* and *K. pneumoniae*, using a conservative approach. Specifically, the MIC values of FTIs were set to the highest tested concentrations, as no inhibitory effect was observed without the simultaneous use of colistin in gram-negative strains. This conservative methodology likely underestimates the true extent of synergy, as higher individual MIC values for FTIs would yield even more pronounced synergistic FICI values. However, the approach used in this study aligns with previous studies that assessed FICI values for substances exhibiting no standalone effect on gram-negative bacteria, such as baicalin, flavomycin, and resveratrol [10,82].

Beyond the already demonstrated potential of FTIs as antimicrobial agents against gram-positive and gram-negative bacteria, this study also highlights their ability to reduce the MIC of colistin for colistin-resistant bacteria. As the prevalence of colistin-resistant strains continues to rise, colistin – once discontinued due to its neurotoxicity and nephrotoxicity – has been reinstated as a "last-resort" antibiotic for treating infections caused by gram-negative nosocomial pathogens such as *A. baumannii* [83]. The findings of this study may have broader implications for addressing colistin-resistant bacteria. To evaluate this hypothesis, further testing on diverse colistin-resistant strains, including those harboring mobile colistin resistance genes, is warranted. Additionally, this approach could target bacteria with a strong propensity for heteroresistance to colistin and other polymyxins. Although heteroresistance can occur in both gram-positive and gram-negative bacteria, its increasing prevalence in gram-negative pathogens such as *E. coli* and *K. pneumoniae* is of particular concern [84]. In this study, the tendency of *K. pneumoniae* to form subcolonies appeared to be diminished when tipifarnib was combined with colistin, suggesting that tipifarnib may contribute to reducing the MIC of colistin and potentially affect heteroresistance in this strain. Further research should be conducted to determine whether the observed inhibition patterns are attributable to the modulation of heteroresistance or rather reflect compound-specific characteristics [85].

For any potential therapeutic use of FTIs as antimicrobial agents in humans, whether as adjuvants or monotherapies, *in vivo* concentrations would need to approach MIC values. For gram-negative strains, the highest FICI values were observed at FTI concentrations of 15.6 µM to 31.3 µM, although even lower concentrations, such as 7.8 µM, demonstrated a reduction in colistin MIC. For example, in a study involving patients with advanced solid tumors, lonafarnib achieved a maximum plasma concentration of 3.8 µg/ml after administering 300 mg twice daily [86], equivalent to approximately 6.0 µM. While such high doses of an anticancer drug are unlikely to be used for treating infections, this scenario might be feasible in cases of severe healthcare-associated infections. Beyond our *in vitro* findings, animal models could provide a more realistic representation of both the pharmacokinetics and pharmacodynamics of FTIs, offering valuable insights into their *in vivo* efficacy and antimicrobial potential. Future studies should incorporate diverse testing media and preclinical models to validate and build upon the results observed *in vitro*. These approaches could also elucidate the impact of FTIs on the human gut microbiome – an important consideration for systemic use – given its potential relevance in cancer therapy [87]. It should also be considered that bacteria may metabolize FTIs, although the specific metabolites remain unknown. Furthermore, even if systemic plasma concentrations remain insufficient, FTIs could have potential as topical agents, allowing for higher localized concentrations in specific areas, as has been shown for statins before [64].

## Conclusion

In summary, this basic research explored the effects of FTIs and additional compounds on key pathogenic bacterial strains in combination with sub-inhibitory concentrations of colistin. The non-peptidomimetic FTIs, lonafarnib and tipifarnib, demonstrated significant activity against gram-negative bacteria when co-administered with sub-inhibitory concentrations of colistin, extending their activity beyond gram-positive bacteria. Notably, FTIs were able to reduce colistin MICs, even in a colistin-resistant gram-negative strain. Peptidomimetic FTIs, B581 and FTI-277, were also effective against gram-negative bacteria but exhibited no activity against gram-positive strains. In contrast, αHFP and bempedoic acid showed no effect on any tested strains, supporting the hypothesis that the mechanism of action of FTIs differs substantially from their activity in eukaryotic cells. The observed effects are likely due to the disruption of multiple biosynthetic pathways, underscoring the potential of FTIs as adjunctive drugs in antimicrobial therapy. Further studies will be conducted to clarify the mechanisms and potential of FTIs as pleiotropic antimicrobial substances, potentially making a leap in combating the ever-increasing threat of resistant pathogens.

## Supporting information

**S1 File. Additional information and controls.** The file provides supporting information on the Tecan device and the program used for fluorescence measurements, as well as details of the conducted controls.
(DOCX)

**S1 Fig. RFI values for *A. baumannii*.** Relative fluorescence intensity (RFI) values for all checkerboard assays with colistin and the six tested substances for *A. baumannii*. Minimum bactericidal concentrations (MBCs) are indicated in bold and italic. All values were normalized to the fluorescence of the corresponding growth control to ensure comparability across plates. All checkerboard assays were conducted in triplicate.
(TIF)

**S2 Fig. RFI values for *B. subtilis*.** Relative fluorescence intensity (RFI) values for all checkerboard assays with colistin and the six tested substances for *B. subtilis*. Minimum bactericidal concentrations (MBCs) are indicated in bold and italic. All values were normalized to the fluorescence of the corresponding growth control to ensure comparability across plates. All checkerboard assays were conducted in triplicate.
(TIF)

**S3 Fig. RFI values for *E. cloacae*.** Relative fluorescence intensity (RFI) values for all checkerboard assays with colistin and the six tested substances for *E. cloacae*. Minimum bactericidal concentrations (MBCs) are indicated in bold and italic. All values were normalized to the fluorescence of the corresponding growth control to ensure comparability across plates. All checkerboard assays were conducted in triplicate.
(TIF)

**S4 Fig. RFI values for *E. faecium*.** Relative fluorescence intensity (RFI) values for all checkerboard assays with colistin and the six tested substances for *E. faecium* Minimum bactericidal concentrations (MBCs) are indicated in bold and italic. All values were normalized to the fluorescence of the corresponding growth control to ensure comparability across plates. All checkerboard assays were conducted in triplicate.
(TIF)

**S5 Fig. RFI values for *E. coli*.** Relative fluorescence intensity (RFI) values for all checkerboard assays with colistin and the six tested substances for *E. coli*. Minimum bactericidal concentrations (MBCs) are indicated in bold and italic. All values were normalized to the fluorescence of the corresponding growth control to ensure comparability across plates. All checkerboard assays were conducted in triplicate.
(TIF)

**S6 Fig. RFI values for *K. pneumoniae*.** Relative fluorescence intensity (RFI) values for all checkerboard assays with colistin and the six tested substances for *K. pneumoniae*. Minimum bactericidal concentrations (MBCs) are indicated in bold and italic. All values were normalized to the fluorescence of the corresponding growth control to ensure comparability across plates. All checkerboard assays were conducted in triplicate.
(TIF)

**S7 Fig. RFI values for MRSA.** Relative fluorescence intensity (RFI) values for all checkerboard assays with colistin and the six tested substances for MRSA. Minimum bactericidal concentrations (MBCs) are indicated in bold and italic. All values were normalized to the fluorescence of the corresponding growth control to ensure comparability across plates. All checkerboard assays were conducted in triplicate.
(TIF)

**S8 Fig. RFI values for *P. paraeruginosa*.** Relative fluorescence intensity (RFI) values for all checkerboard assays with colistin and the six tested substances for *P. paraeruginosa*. Minimum bactericidal concentrations (MBCs) are indicated in bold and italic. All values were normalized to the fluorescence of the corresponding growth control to ensure comparability across plates. All checkerboard assays were conducted in triplicate.
(TIF)

**S9 Fig. RFI values for *S. aureus*.** Relative fluorescence intensity (RFI) values for all checkerboard assays with colistin and the six tested substances for *S. aureus*. Minimum bactericidal concentrations (MBCs) are indicated in bold and italic. All values were normalized to the fluorescence of the corresponding growth control to ensure comparability across plates. All checkerboard assays were conducted in triplicate.
(TIF)

**S10 Fig. RFI values for *S. epidermidis*.** Relative fluorescence intensity (RFI) values for all checkerboard assays with colistin and the six tested substances for *S. epidermidis*. Minimum bactericidal concentrations (MBCs) are indicated in bold and italic. All values were normalized to the fluorescence of the corresponding growth control to ensure comparability across plates. All checkerboard assays were conducted in triplicate.
(TIF)

## Acknowledgments

The authors would like to thank Tim Bürgel for his advice and introduction to image creation with BioRender and Raffaela Maltaner for her technical support in the laboratory.

## Author contributions

**Conceptualization:** Marian Klose, Lea Weber, Hagen Sjard Bachmann.

**Data curation:** Marian Klose.

**Formal analysis:** Marian Klose, Lea Weber.

**Investigation:** Marian Klose.

**Methodology:** Marian Klose.

**Project administration:** Hagen Sjard Bachmann.

**Supervision:** Hagen Sjard Bachmann.

**Validation:** Lea Weber, Hagen Sjard Bachmann.

**Visualization:** Marian Klose, Lea Weber.

**Writing – original draft:** Marian Klose.

**Writing – review & editing:** Marian Klose, Lea Weber, Hagen Sjard Bachmann.

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
