## [Decision Letter · Decision Letter 0]

16 Jun 2025

In Vitro Synergy of Farnesyltransferase Inhibitors in Combination with Colistin against ESKAPE Bacteria

PLOS ONE

Dear Dr. Bachmann,

Thank you for submitting your manuscript to PLOS ONE. After careful consideration, we feel that it has merit but does not fully meet PLOS ONE’s publication criteria as it currently stands. Therefore, we invite you to submit a revised version of the manuscript that addresses the points raised during the review process.

We look forward to receiving your revised manuscript.

Kind regards,

Shengwei Sun, Ph.D.

Academic Editor

PLOS ONE

Journal Requirements:

2. We note that the S1_File in your submission contains a copyrighted image. All PLOS content is published under the Creative Commons Attribution License (CC BY 4.0), which means that the manuscript, images, and Supporting Information files will be freely available online, and any third party is permitted to access, download, copy, distribute, and use these materials in any way, even commercially, with proper attribution. For more information, see our copyright guidelines: http://journals.plos.org/plosone/s/licenses-and-copyright.

 1. You may seek permission from the original copyright holder of the Figure in S1_File to publish the content specifically under the CC BY 4.0 license.

Reviewers' comments:

Reviewer's Responses to Questions

**Comments to the Author**

1. Is the manuscript technically sound, and do the data support the conclusions?

Reviewer #1: Yes

2. Has the statistical analysis been performed appropriately and rigorously?

Reviewer #1: Yes

3. Have the authors made all data underlying the findings in their manuscript fully available?

Reviewer #1: Yes

4. Is the manuscript presented in an intelligible fashion and written in standard English?

Reviewer #1: Yes

Reviewer #1: The manuscript investigates the effects of FTIs and additional compounds on key pathogenic bacteria when co-administered with colistin. The potential synergistic interactions were illustrated by a modified checkerboard assay. The approach enables FTIs to exhibit antimicrobial activity against several gram-negative bacteria when co-administered with sub-inhibitory concentrations of colistin. The authors found FTIs were able to reduce colistin MICs, even in a colistin-resistant gram-negative strain, thus offering a novel strategy to combat antimicrobial resistance. However, several key issues are not clarified clearly, some modification should be made according to the following questions:

Comments:

1. Line 216, why does S. aureus MRSA have antibiotic resistance to the quantities of ampicillin, chloramphenicol, kanamycin and tetracycline, while S. aureus doesn’t?

2. Line 223, the author should provide the specific value of the sub-inhibitory concentration of colistin.

3. In the “Growth kinetics” section, the author only conducted the growth kinetics for S. aureus. Why did the authors not monitor the dynamic changes in the combined inhibitory effects of FTIs and colistin on E. cloacae growth? Time-kill assays should be included to evaluate the antibacterial effect.

4. Line 382, the author should discuss the lack of activity of the peptidomimetic FTIs, B581 and FTI-277 against gram-positive bacteria. Why the antimicrobial effects of peptidomimetic FTIs lower than non-peptidomimetic FTIs?

5. Line 438-440 and figure 8, the author only speculated the mechanism of the synergistic effects on gram-negative bacteria with the combined use of colistin and FTIs by disrupting the OM’s integrity and increasing permeability. The conclusion is not supported by any data. To further strengthen the conclusions, the authors may consider performing outer membrane permeability assay to confirm.

6. Line 476-479, “…colistin-resistant strains, including those harboring mobile colistin resistance genes “, “this approach could target bacteria with a strong propensity for heteroresistance to colistin and other polymyxins”, please give examples to illustrate which bacteria are included respectively.

7. Line 481-483, could you give more examples/details about how the FTIs affect the heteroresistance in gram-negative pathogens bacteria with or without colistin?

**Do you want your identity to be public for this peer review?** For information about this choice, including consent withdrawal, please see our Privacy Policy

Reviewer #1: No

---

## [Author Response · Author response to Decision Letter 1]

30 Jul 2025

Dear Editor, dear Reviewer,

We would like to sincerely thank you for your thorough evaluation of our manuscript and the opportunity to contribute to PLOS ONE. We greatly appreciate the constructive feedback and the chance to further improve our work in response to your valuable comments.

We have carefully reviewed all submitted files and, to the best of our knowledge, ensured that they now fully comply with PLOS ONE's style and file naming requirements. Concerning the use of the image in the S1_File, we contacted the Tecan Deutschland GmbH, which provides both the device and the associated software shown in the screenshot. We received written confirmation that the reported screenshot does not constitute a copyright issue and can be used within the scope of this publication without permission. The relevant correspondence is provided as a supporting document.

We now respond point-by-point to the reviewer’s comments below.

1. Line 216, why does S. aureus MRSA have antibiotic resistance to the quantities of ampicillin, chloramphenicol, kanamycin and tetracycline, while S. aureus doesn’t?

We agree that this aspect needed clarification. The MRSA strain used in our study, Staphylococcus aureus ATCC 33592, is mecA-positive (PMID: 19752281), whereas S. aureus ATCC 25923 is mecA-negative and widely used as a quality control strain (PMID: 25377701). The methicillin-sensitive strain is markedly more susceptible to the antibiotics tested compared to the methicillin-resistant one. It is well established that methicillin resistance in S. aureus is frequently associated with cross-resistances, and thus often coincides with multidrug resistance (PMID: 32257966). To demonstrate that established antibiotics can still be effective against this MRSA strain, we additionally included gentamicin and vancomycin in our control experiments, as shown in Fig 3.

2. Line 223, the author should provide the specific value of the sub-inhibitory concentration of colistin.

The exact concentrations of colistin used in the combination experiments are already detailed in the Materials and Methods section under the description of the checkerboard assay, and are also indicated in the respective figures. Nevertheless, in accordance with the reviewer’s recommendation, we have now included the concentration range directly in the Results section for clarity (Line 248-250).

3. In the “Growth kinetics” section, the author only conducted the growth kinetics for S. aureus. Why did the authors not monitor the dynamic changes in the combined inhibitory effects of FTIs and colistin on E. cloacae growth? Time-kill assays should be included to evaluate the antibacterial effect.

This is a valid point, and we have now extended our analysis by conducting additional growth kinetics for E. cloacae, specifically focusing on the combined application of colistin and FTIs to illustrate potential synergistic effects. Furthermore, we have implemented time-kill assays for both S. aureus and E. cloacae to more thoroughly assess the bactericidal activity and the dynamic interplay between colistin and FTIs over time. These new experiments have been incorporated into the revised manuscript (Line 33, 185-192, 210-213, 340, and 351-390).

4. Line 382, the author should discuss the lack of activity of the peptidomimetic FTIs, B581 and FTI-277 against gram-positive bacteria. Why the antimicrobial effects of peptidomimetic FTIs lower than non-peptidomimetic FTIs?

The differences in antimicrobial activity between the tested FTIs likely result from distinct chemical properties. Lonafarnib and tipifarnib, which affect gram-positive and gram-negative bacteria, both carry halogen substituents (bromine and chlorine, respectively). Halogens are known to be important for the antibiotic activity of different classes of antibiotics, like chloramphenicol, tetracyclines, and fluoroquinolones. Faleye et al. (2023; PMID: 37845080) emphasized that halogenated compounds often show increased affinity for gram-positive bacteria. Similarly, Sanabria-Rios et al. (2022; PMID: 36118101) demonstrated brominated fatty acids to be effective against MRSA strains. These findings may explain the higher activity of lonafarnib and tipifarnib. In contrast, the peptidomimetic FTIs B581 and FTI-277 contain polar thiol groups, which may impair membrane permeability. We have added an explanation of the chemical structures and properties of the molecules to the manuscript (Line 471-475). Further investigations into membrane integrity and compound uptake will help clarify the role of these functional groups in antimicrobial activity.

5. Line 438-440 and figure 8, the author only speculated the mechanism of the synergistic effects on gram-negative bacteria with the combined use of colistin and FTIs by disrupting the OM’s integrity and increasing permeability. The conclusion is not supported by any data. To further strengthen the conclusions, the authors may consider performing outer membrane permeability assay to confirm.

To address this concern, we conducted outer membrane permeability assays using the fluorescent probe N-Phenyl-1-naphthylamine (NPN) for E. coli as a model strain and E. cloacae, a colistin-resistant isolate, as representative gram-negative bacteria. The results confirmed that sub-inhibitory concentrations of colistin were sufficient to increase outer membrane permeability without affecting viability. These new data have been included in the revised manuscript (Line 194-204, 392-415).

6. Line 476-479, “…colistin-resistant strains, including those harboring mobile colistin resistance genes “, “this approach could target bacteria with a strong propensity for heteroresistance to colistin and other polymyxins”, please give examples to illustrate which bacteria are included respectively.

As we were able to demonstrate a significant antimicrobial effect of FTIs against gram-negative bacteria when co-administered with sub-inhibitory concentrations of colistin, the required colistin concentration was notably reduced. This finding suggests that even colistin-resistant bacteria might be susceptible to such combination treatments. Notably, one of the tested strains, E. cloacae ATCC 13047, is known to be highly resistant to colistin (PMID: 20207761). Members of the Enterobacter cloacae complex (ECC) are not only important nosocomial pathogens but are also well documented for their high potential to exhibit colistin heteroresistance (PMID: 36627272). Furthermore, plasmid-mediated colistin resistance genes (mcr) have been identified across several gram-negative ESKAPE organisms, including E. coli (PMID: 30305321), K. pneumoniae (PMID: 34946038), and A. baumannii (PMID: 30936095). While we did not include mcr-positive isolates in our study, future work will investigate whether FTIs may help restore susceptibility in strains carrying mcr-mediated colistin resistance. Notably, previous studies have shown that mcr-mediated colistin resistance can be overcome by using rationally designed combination therapies (PMID: 29386620). These findings underscore the potential of similar approaches – such as combining FTIs with sub-inhibitory colistin – to restore susceptibility in resistant gram-negative pathogens.

7. Line 481-483, could you give more examples/details about how the FTIs affect the heteroresistance in gram-negative pathogens bacteria with or without colistin?

This point raises an important aspect of our findings. When used alone, FTIs showed no impact on most gram-negative bacteria, clearly indicating resistance rather than heteroresistance. In our study, we mentioned the possibility that FTIs in combination with colistin potentially influence the formation of heteroresistant subcolonies. For instance, lonafarnib, when combined with sub-inhibitory colistin, did not consistently suppress bacterial growth across the triplicate near the MIC threshold in the K. pneumoniae checkerboard assay, which supports the hypothesis of heteroresistant escape events. However, this remains a hypothesis. We have therefore deleted the word 'heteroresistant' from the Results section and now describe the formation of subcolonies only.

However, we assume that the observed inconsistencies in growth inhibition patterns, particularly near the MIC threshold, are characteristic of heteroresistance, where subpopulations can transiently survive otherwise inhibitory concentrations (PMID: 25567227). We therefore proceed to discuss this hypothesis in greater detail in the Discussion (Line 574-576).

---

## [Editor Report · Decision Letter 1]

18 Aug 2025

In Vitro Synergy of Farnesyltransferase Inhibitors in Combination with Colistin against ESKAPE Bacteria

PONE-D-25-17781R1

Dear Dr. Bachmann,

We’re pleased to inform you that your manuscript has been judged scientifically suitable for publication and will be formally accepted for publication once it meets all outstanding technical requirements.

Kind regards,

Shengwei Sun, Ph.D.

Academic Editor

PLOS ONE
---

## [Editor Report · Acceptance letter]

PONE-D-25-17781R1

PLOS ONE

Dear Dr. Bachmann,

I'm pleased to inform you that your manuscript has been deemed suitable for publication in PLOS ONE. Congratulations! Your manuscript is now being handed over to our production team.

Kind regards,

on behalf of

Dr. Shengwei Sun

Academic Editor

PLOS ONE